# Effect of Pefloxacin on *Clostridioides difficile* R20291 Persister Cells Formation

**DOI:** 10.3390/antibiotics14070628

**Published:** 2025-06-20

**Authors:** Camila Queraltó, Iván L. Calderón, Isidora Flores, José Rodríguez, Osvaldo Inostroza, Ruth González, Daniel Paredes-Sabja, Jorge A. Soto, Juan A. Fuentes, Fernando Gil

**Affiliations:** 1Doctorado en Biociencias Moleculares, Facultad de Ciencias de la Vida, Universidad Andres Bello, Santiago 8370186, Chile; camilaabeleen25@gmail.com; 2Microbiota-Host Interactions & Clostridia Research Group, Center for Biomedical Research and Innovation (CIIB), Universidad de los Andes, Santiago 7620001, Chile; isidoranatalia16@gmail.com (I.F.); jose.99.rb@gmail.com (J.R.); osv.inostroza.t@gmail.com (O.I.); 3Laboratorio de RNAs Bacterianos, Facultad de Ciencias de la Vida, Universidad Andres Bello, República 330, Santiago 8370186, Chile; lcalderon@unab.cl; 4Departamento de Ciencias Biológicas, Facultad de Ciencias de la Vida, Universidad Andres Bello, República 330, Santiago 8370186, Chile; ruth.gonzalez@unab.cl; 5Interdisciplinary Program in Genetics & Genomics, Texas A&M University, College Station, TX 77840, USA; dparedes-sabja@bio.tamu.edu; 6Department of Biology, Texas A&M University, College Station, TX 77840, USA; 7ANID-Millennium Science Initiative Program-Millennium Nucleus in the Biology of the Intestinal Microbiota, Santiago 8370186, Chile; 8Laboratorio de Inmunología Traslacional, Centro de Investigación de Resiliencia a Pandemias, Millennium Institute on Immunology and Immunotherapy, Facultad de Ciencias de la Vida, Universidad Andres Bello, Santiago 8370186, Chile; jorge.soto.r@unab.cl; 9Laboratorio de Genética y Patogénesis Bacteriana, Centro de Investigación de Resiliencia a Pandemias, Facultad de Ciencias de la Vida, Universidad Andres Bello, República 330, Santiago 8370186, Chile; 10School of Medicine, Faculty of Medicine, Universidad de los Andes, Santiago 7620001, Chile

**Keywords:** persister cells, *Clostridioides difficile*, pefloxacin

## Abstract

*Clostridioides difficile* is a Gram-positive bacterium recognized for its ability to produce toxins and form spores. It is mainly accountable for the majority of instances of antibiotic-related diarrhea. **Background.** Bacterial persister represent a minor fraction of the population that shows temporary tolerance to bactericidal agents, and they pose considerable medical issues because of their link to the rise of antibiotic resistance and challenging chronic or recurrent infections. Our previous research has shown a persister-like phenotype associated with treatments that include pefloxacin. Nonetheless, the mechanism is still mostly unclear, mainly because of the difficulty in isolating this small group of cells. **Objectives.** To enhance the understanding of *C. difficile* persister cells, we made an enrichment and characterization of these cells from bacterial cultures during the exponential phase under pefloxacin treatment and lysis treatment. **Results.** We demonstrate the appearance of cells with lower metabolism and DNA damage. Furthermore, we noted the participation of toxin–antitoxin systems and Clp proteases in the generation of persister cells. **Conclusions.** This work demonstrates the formation of *C. difficile* persister cells triggered by a lethal concentration of pefloxacin.

## 1. Introduction

*Clostridioides difficile* (*C. difficile*) is a Gram-positive, strictly anaerobic, spore-forming bacterium capable of colonizing and proliferating within the human gastrointestinal tract [1]. It is recognized as the leading cause of antibiotic-associated diarrhea, accounting for approximately 30% of cases in hospital settings [2]. The clinical manifestations of *C. difficile* infection (CDI) vary widely, ranging from mild diarrhea to severe conditions such as pseudomembranous colitis, toxic megacolon, sepsis, and even death [3]. Mortality rates associated with CDI range from approximately 5.7% during endemic periods up to 19% during epidemic outbreaks [4,5]. A major clinical challenge posed by CDI is its high recurrence rate, which increases progressively from about 30% after the initial episode to 45% and 64% following the second and third episodes, respectively [6].

The high recurrence rate is closely associated with the ability of *C. difficile* to undergo sporulation, a survival mechanism triggered by unfavorable environmental conditions. These spores exhibit resistance to standard antibiotic therapies and adhere to the colonic epithelium [7]. Consequently, upon cessation of antibiotic treatment, spores can germinate and initiate reinfection, thereby perpetuating recurrent episodes [7]. Although spores are widely accepted as the primary morphotype responsible for CDI persistence, recent murine studies suggest additional biological factors may also play critical roles [8,9].

Among these factors, mechanisms involved in bacterial adherence and biofilm formation have attracted considerable attention due to their contribution to infection persistence and antibiotic tolerance [10,11]. Biofilms confer increased resistance to clinically relevant antibiotics, such as vancomycin and metronidazole, thus complicating treatment regimens [10,11]. Moreover, intestinal biofilms have been identified as reservoirs that promote recurrent infections, highlighting their crucial role in both colonization and persistence of *C. difficile* [6,12].

Beyond biofilms, persister cells have emerged as significant contributors to recurrent bacterial infections. Persister cells represent a subpopulation of dormant bacterial cells characterized by low metabolic activity, enabling them to survive prolonged exposure to bactericidal antibiotics without replication [13,14,15]. Their formation has profound clinical implications, not only facilitating recurrence but also contributing to therapeutic failures and difficulties in eradicating infections despite appropriate antibiotic regimens [16,17,18,19]. In the context of CDI, such persister cells could play a pivotal role in maintaining infection persistence and facilitating subsequent reactivation of disease.

Persister cells are broadly classified into two types: type I (stochastic persisters), which arise spontaneously during bacterial growth, and type II (triggered persisters), which form in response to specific environmental stimuli [20,21,22]. While the exact mechanisms governing type II persister cell formation remain incompletely understood, antibiotic exposure has been consistently identified as a critical inducing factor. Furthermore, the expression patterns of persister-related genes vary depending on the nature of the inducing stimulus [21,22,23,24,25]. Specifically, the emergence of persister cells following treatment with fluoroquinolones, a class of antibiotics known to inhibit bacterial DNA gyrase and topoisomerase IV, thereby inducing DNA damage, has been well documented [26,27,28,29].

In organisms such as *Staphylococcus aureus* and *Pseudomonas aeruginosa*, fluoroquinolone exposure promotes high levels of persister formation via the bacterial SOS response pathway [27]. In *Escherichia coli*, ciprofloxacin-induced DNA damage activates RecA-mediated cleavage of the LexA repressor, consequently upregulating the SOS response and enhancing persister cell formation with increased tolerance to fluoroquinolones and β-lactam antibiotics [26,27]. Additionally, ciprofloxacin treatment increases persister cell formation within biofilms by modulating type II toxin–antitoxin (TA) systems in *P. aeruginosa* [28].

Preliminary studies from our laboratory indicate that exposure to pefloxacin, a fluoroquinolone antibiotic, may also induce persister cell formation in *C. difficile*; however, this phenomenon has not yet been thoroughly investigated [30,31]. Although previous research has demonstrated the capacity of *C. difficile* to form both type I and type II persister cells, notably in response to vancomycin, a commonly employed antibiotic for CDI treatment, there is limited knowledge regarding the influence of fluoroquinolone antibiotics like pefloxacin on persister formation in this pathogen [31].

Considering the potential exposure of *C. difficile* to fluoroquinolones in the gut environment, particularly in asymptomatic carriers undergoing antibiotic treatment, investigating the impact of pefloxacin on persister cell induction is clinically relevant. Fluoroquinolones could thus significantly affect spore germination, infection development, and persistence dynamics in early infection stages [32,33].

Persistent bacterial infections pose a critical clinical challenge due to the ability of certain bacterial subpopulations (the persister cells) to transiently tolerate lethal antibiotic concentrations without undergoing genetic mutations conferring resistance. Unlike resistant cells, persisters rely on phenotypic switching to a dormant, metabolically quiescent state, making their eradication particularly challenging and leading to recurrent infections despite seemingly successful antibiotic treatments. Understanding the molecular mechanisms underlying persister formation, especially in pathogens with high clinical relevance, such as *C. difficile*, is therefore essential for developing more effective therapeutic strategies.

In the specific context of CDI, antibiotics play a dual and paradoxical role. While necessary for treating the infection, certain antibiotic classes disrupt the healthy intestinal microbiota, inadvertently promoting spore germination, colonization, and disease recurrence. Other studies have reported that fluoroquinolones, particularly pefloxacin, similarly contribute to persister formation in diverse bacterial species, including *C. difficile* itself, underscoring the complexity and breadth of antibiotic-induced persistence mechanisms. Despite these findings, the detailed molecular processes driving fluoroquinolone-induced persistence in *C. difficile* remain poorly understood.

Addressing this knowledge gap, the present study provides significant advancements in comprehending persister formation dynamics specifically triggered by pefloxacin in *C. difficile*. By employing a robust experimental framework combining antibiotic exposure, selective lysis enrichment, and precise phenotypic characterization via flow cytometry, we conclusively identified and isolated metabolically quiescent, viable persister cells. Unlike previous studies that primarily characterized general phenotypic aspects, our approach deeply explores the underlying cellular and molecular responses, particularly DNA damage, SOS induction, peptidase–chaperone interactions, and toxin–antitoxin (TA) system activation, that govern persister formation in response to pefloxacin.

## 2. Results

### 2.1. Pefloxacin Induces Biphasic Kill Kinetics and a Persistent Subpopulation in C. difficile

To investigate whether pefloxacin triggers persister cell formation in *C. difficile*, we first determined the minimum inhibitory concentration (MIC) of pefloxacin for strain R20291, which was 1.53 µg/mL. Exponential-phase cultures were then challenged with pefloxacin (Pef) at 1×, 10×, 50×, and 100× MIC. As shown in Figure 1A, exposure to 1× MIC allowed growth, with CFU counts rising from 10^8^ to approximately 10^9^ CFU/mL over 6 h. Treatment at 10× MIC resulted in a gradual decline to ~10^7^ CFU/mL by 6 h without an apparent plateau. In contrast, 50× MIC induced a characteristic biphasic killing curve with statistically different slopes (see Appendix A). In this high-dose treatment, an initial rapid reduction in viable cells occurred during the first 2 h, followed by a persistent plateau at ~10^5^–10^6^ CFU/mL lasting until the 6 h endpoint, indicative of a non-replicating persister subpopulation. For 100× MIC the biphasic curve was not clear at 6 h. For longer time (8 h) see Appendix A and Appendix A to see non-statistically slope change.

To confirm that the plateau phase reflected actual persistence rather than heritable resistance, survivors from 50× MIC treatments were collected at 6 h, washed, resuspended in fresh brain heart infusion supplemented with taurocholate, fructose, and glucose (++ conditions), and re-exposed to pefloxacin at 50× MIC. As depicted in Figure 1B, the second challenge reproduced the biphasic kill profile, and subsequent MIC testing remained unchanged (1.53 µg/mL), confirming the retention of antibiotic susceptibility and reversible dormancy. Moreover, a Δ*spo0A* mutant, defective for sporulation, exhibited identical biphasic survival kinetics under 50× MIC treatment, demonstrating that spore formation did not contribute to the observed persistence. Collectively, these findings establish that high-concentration pefloxacin exposure induces a reversible, non-sporulating persister cell subpopulation in *C. difficile*.

To distinguish between stochastic and antibiotic-triggered persisters, we subjected exponential-phase cultures of *C. difficile* R20291 to a 15 min lysis treatment either without antibiotic exposure (Figure 2A) or after a 50× MIC pefloxacin challenge (Figure 2B), as previously described [31,34]. In the absence of the antibiotic, lysis of untreated cultures resulted in a low and time-independent survival rate of approximately 1.8%, reflecting the basal frequency of stochastic (type I) persister cells. By contrast, when cultures were pre-exposed to pefloxacin for 30 min, 1, 2, and 4 h before lysis, survival rose markedly, peaking at ~10% after 2 h and remaining stable through 4 h. This difference between antibiotic-treated and untreated lysed samples was highly significant (*p* < 0.005). Notably, the persistence profiles of the wild-type and Δ*spo0A* mutant strains (black and gray bars, respectively) were indistinguishable at all time points (ns), thereby excluding a role for *Spo0A*-mediated sporulation in persister enrichment. Together, these data confirm that high-dose pefloxacin specifically induces a non-sporulating persister subpopulation in *C. difficile* R20291, which can be selectively enriched through lysis treatment.

Persister cells are characterized by metabolic dormancy and maintained viability [13,35,36]. To probe these features, we employed thioflavin-T (Thio-T) as an RNA-binding fluorescent probe and propidium iodide (PI) as a viability indicator [37]. Thio-T selectively intercalates into structured RNA molecules, with fluorescence intensity correlating to RNA content and metabolic activity. In contrast, PI is impermeant to cells with intact membranes but penetrates cells with compromised membranes, where it binds DNA and fluoresces red, signaling loss of membrane integrity.

In untreated control cultures, the vast majority of cells exhibited intense Thio-T fluorescence and negligible PI uptake (Figure 3, control, row 1), confirming robust RNA synthesis and intact cell envelopes. When cultures were treated with pefloxacin alone (PEF; Figure 3, PEF, row 3), Thio-T labeling decreased markedly, and a small fraction of cells became PI-positive, indicating that antibiotic stress impairs RNA metabolism and induces limited membrane damage. Lysis buffer treatment without prior antibiotic exposure (Figure 3, TL, row 4) resulted in near-absence of Thio-T signal and no detectable PI fluorescence, consistent with extensive membrane disruption and depletion of intact RNA.

Strikingly, the combined pefloxacin-plus-lysis protocol (Figure 3, PEF + LT, row 2) yielded a distinct subpopulation characterized by strong Thio-T positivity and minimal PI staining. Moreover, heterogeneity in Thio-T intensity among these survivors suggested variable, but overall attenuated, RNA synthesis rates, hallmarks of persister cell physiology. These observations demonstrate that the lysis-enriched fraction from pefloxacin-treated cultures comprises viable cells with preserved membrane integrity yet reduced metabolic activity, suggesting their identity as bona fide persisters in *C. difficile* R20291.

### 2.2. Flow Cytometric Analysis of Persister-Associated Metabolic and Viability States

To quantify the distribution of metabolic and membrane-intact subpopulations at the single-cell level, we performed flow cytometry on dual-labeled cultures (Thio-T and PI), as described [31,34,37]. Representative dot plots are shown in Figure 4 left, with quadrants defining Thio-T+ only (upper-left), double-positive (DP; upper-right), double-negative (DN; lower-left), and PI+ only (lower-right) cells.

Flow cytometric analysis of dual-labeled *C. difficile* cultures revealed treatment-dependent shifts in metabolic activity (Thio-T) and membrane integrity (PI) subpopulations [31,34] (Figure 4). To identify which treatment most effectively enriches bona fide persister cells, operationally defined as those that survive antibiotic challenge with intact membranes (PI^−^) and retained metabolic activity (Thio-T^+^), we compared the four conditions using both Figure 4 and Appendix A. Under the combined pefloxacin + lysis protocol (PEF + TL), the Thio-T^+^ only fraction remained high at 83.5%, statistically indistinguishable from the untreated control (84.7%; ns), while PI^+^ cells stayed low at 1.8% (ns). Double-positive (DP) and double-negative (DN) populations under PEF + TL likewise did not differ from control (4.6% DP, 10.0% DN; all ns) (Figure 4 left, Appendix A).

By contrast, pefloxacin alone (PEF) modestly reduced Thio-T^+^ cells to 79.8% and increased PI^+^ to 2.5%, and lysis alone (TL) decreased Thio-T^+^ to 53.4% (compared with CT) while elevating DN to 42.2% (compared with CT) (Figure 4 left; Appendix A). The considerable DN rise under TL reflects extensive metabolic shutdown rather than selective antibiotic tolerance, and the left-shifted Thio-T histogram confirms loss of activity (Figure 4 right). Taken together, only PEF + LT preserves a high-activity, intact-membrane phenotype matching untreated cells; this balance (high Thio-T^+^/low PI^+^ with control-level DP and DN) suggests the most significant enrichment of true persister cells.

### 2.3. Evaluation of DNA Damage in Pefloxacin-Induced Persister Cells in C. difficile

As a member of the fluoroquinolone class, pefloxacin exerts bactericidal activity via inhibition of topoisomerase IV in Gram-positive organisms and topoisomerase II in Gram-negative species, leading to DNA cleavage complexes and strand breaks [29]. To determine whether pefloxacin-induced DNA damage contributes to persister formation, we performed alkaline comet assays on *C. difficile* cultures enriched for persisters at 0, 15, and 30 min after exposure to 50× MIC pefloxacin, as previously described [38]. The comet assay provides single-cell resolution of genomic integrity, with tail length directly correlating to lesion burden [38,39]. In untreated controls, comets were minimal (<10 µm), confirming baseline genomic integrity (Figure 5, lower panel, No treatment). Conversely, UV-irradiated cells (positive control) displayed extensive DNA fragmentation. In persister-enriched cultures (PEF + TL; top panel), mean tail length rose from <10 µm at 0 min to ~60 µm at 15 min and reached ~154 µm by 30 min. Bulk cultures treated with pefloxacin alone (PEF; middle panel) also accumulated strand breaks, but to a lesser extent (~30 µm at 15 min and ~80 µm at 30 min).

These results demonstrate that high-dose pefloxacin induces time-dependent DNA damage in both total (PEF) and persister-enriched *C. difficile* populations (PEF + TL). The pronounced increase in comet tail length within persister fractions indicates that survival under the pefloxacin challenge does not preclude significant genomic lesions, implicating DNA damage as a hallmark of persister physiology.

### 2.4. Persistence-Related Gene Expression Analysis

Persister formation in bacterial populations is a phenotypic switch governed by tightly regulated stress responses rather than genetic mutations. In *C. difficile*, exposure to high-dose fluoroquinolones triggers a cascade of events (i.e., protease activation, toxin–antitoxin (TA) induction and DNA-damage repair) that collectively enforce growth arrest and enhance survival [25,40,41]. To dissect the transcriptional underpinnings of pefloxacin-induced persistence and link them to our central aim of identifying key molecular determinants in *C. difficile* persister cells, we quantified the expression of select peptidases (ClpP1, ClpP2, Lon), chaperone-associated proteases (ClpC, ClpX, ClpB), TA toxins (MazF, RelE) and the SOS regulator RecA at early (10 min) and late (30 min) stages post-treatment. This targeted analysis reveals the dynamic regulatory program that sustains the persister phenotype under antibiotic stress [27,42].

Transcriptomic analyses have shown that, although persister and non-persister cells are genetically identical, they display distinct expression patterns characterized by repression of growth-related functions and activation of TA modules and DNA repair systems [42]. Type II TA systems comprise a stable toxin and a labile antitoxin, the latter degraded by proteases such as ClpP and Lon under stress, thereby freeing the toxin to inhibit essential processes and arrest growth [25,40,41]. Likewise, activation of the SOS response via RecA-mediated LexA cleavage induces genes that promote persister formation [25,27]. To pinpoint the molecular determinants of pefloxacin-induced persistence, we quantified mRNA levels of the peptidases ClpP1, ClpP2 and Lon; the chaperone-associated proteases ClpC, ClpX and ClpB; the toxins MazF and RelE; the SOS regulator RecA at early (10 min) and late (30 min) time points following 50× MIC pefloxacin treatment in persister-enriched cells (Figure 6).

In untreated controls (CT), all tested transcripts remained at baseline levels (fold-change ≈ 1), confirming minimal stress in the absence of antibiotics or lysis (Figure 6A–F). Upon combined pefloxacin + lysis treatment (PEF + TL), *clpP2* was the earliest and most robustly induced protease, rising to ~6-fold at 10 min compared to CT, whereas pefloxacin alone (PEF) elicited a ~4-fold increase and lysis alone (TL) only ~2-fold. In contrast, *clpP1* and *lon* transcripts remained essentially unchanged across CT, PEF + TL, PEF, and TL at both time points.

Chaperone-associated ATPases followed a similar pattern: at 10 min, *clpC* increased ~3.8-fold in PEF + TL versus ~2.1-fold in PEF and ~2.4-fold in TL, while *clpX* peaked at ~4.2-fold under PEF + TL but showed only ~2.0-fold induction in PEF and no significant change in TL. By 30 min, *clpC* expression had tapered toward CT levels in all treatments, whereas *clpX* further escalated to ~5.5-fold in PEF + TL and ~3.0-fold in PEF, remaining low in TL. *clpB* expression did not differ significantly from CT under any condition or time (Figure 6B,E).

Toxin–antitoxin transcripts exhibited distinct temporal and treatment-dependent behaviors. At 10 min, *relE* was the most highly induced gene in PEF + TL (~7-fold), compared with ~5.5-fold in PEF and ~2.3-fold in TL; *mazF* rose more modestly (2–3-fold) across PEF + TL, PEF and TL. *recA* surged ~5-fold in both PEF + TL and PEF but remained near CT in TL, indicating SOS activation primarily in antibiotic-exposed cells (Figure 6C). At 30 min, *mazF* became the dominant TA factor in PEF + TL (~8-fold) and PEF (~6-fold), whereas *relE* peaked in TL (~5.5-fold) and declined in PEF + TL (~3-fold) and PEF (~2-fold). *recA* expression, although still elevated over CT, decreased relative to its early peak in both PEF + TL and PEF, consistent with a transient DNA-damage response (Figure 6F).

Together, these data demonstrate that the combined PEF + TL protocol orchestrates a sequential stress-response program (i.e., early *clpP2* and *relE* induction, followed by sustained *clpX* upregulation and late-stage *mazF* expression) distinct from the patterns observed under PEF or TL alone. This dynamic regulatory cascade underpins the establishment and maintenance of the *C. difficile* persister phenotype under high-dose fluoroquinolone stress.

## 3. Discussion

Persister cells represent a phenotypic state in bacterial populations characterized by reversible dormancy, which enables their survival under lethal antibiotic conditions without involving genetic mutations [17,43,44]. In the context of infections caused by *C. difficile*, the formation of persisters poses a significant clinical challenge, complicating treatment regimens and promoting recurrent infections. Thus, a deeper understanding of the molecular mechanisms underlying their formation is essential for developing effective therapeutic strategies [31,45,46].

In this study, we focused on the antibiotic pefloxacin, a fluoroquinolone known to disrupt intestinal microbiota balance and facilitate the establishment of CDI [32,33,47,48]. Previous work by our group demonstrated that pefloxacin exposure can induce persister cell formation in *C. difficile* cultures [30,31]. Given these antecedents, we sought to characterize the cellular and molecular processes involved in pefloxacin-triggered persistence more deeply.

Our experimental approach relied on a robust model where exponential-phase cultures of *C. difficile* were challenged with pefloxacin at concentrations ranging from 1× to 100× MIC. Notably, a biphasic killing curve, typical of persister formation [13], was evident at 50× MIC (Figure 1). For 100× MIC, at longer times, we did not clearly appreciate biphasic behavior. Considering this, we decide to use 50× MIC for all experiments. We subsequently enriched persister populations through selective lysis treatments, confirming through revival experiments that these cells were indeed persisters and not antibiotic-resistant mutants, consistent with previous reports [31,34]. The use of a Δ*spo0A* mutant strain confirmed that the observed phenotypes arose solely from vegetative cells, eliminating potential confusion from spore formation.

The enriched persister phenotype was precisely characterized via flow cytometry employing dual-staining with PI and Thio-T. PI staining identifies cells with compromised membranes, while Thio-T indicates RNA metabolic activity. The population enriched with pefloxacin followed by lysis treatment (PEF + TL) maintained a high proportion of Thio-T-positive and PI-negative cells, thus verifying their viability with reduced metabolic activity, which are hallmarks of persister cells. This approach represents an effective adaptation of methodologies traditionally used in aerobic organisms [34], addressing the limitations associated with *C. difficile*’s anaerobic nature.

Notably, flow cytometric analyses revealed a significant heterogeneity within the persister-enriched populations, highlighting the coexistence of cells at different metabolic states, ranging from low metabolic activity (DN cells) to actively repairing cells (DP cells), reflecting the metabolic diversity known to exist among persister subpopulations [36]. Unlike previous observations with vancomycin, pefloxacin-induced persistence did not cause notable morphological changes, likely due to its distinct mechanism of action involving DNA synthesis inhibition rather than cell-wall disruption [31,49].

To confirm pefloxacin’s DNA-damaging effects as a driver of persistence, we conducted comet assays. Results clearly demonstrated that pefloxacin treatment induced substantial DNA damage, evidenced by progressively elongated comet tails, suggesting that persistence under fluoroquinolone stress involves cells capable of enduring extensive DNA lesions without immediate lethality (Figure 5). Fluoroquinolones target DNA topoisomerase IV, forming stable DNA-enzyme complexes that prevent the ligation of transiently cleaved DNA strands, thus accumulating lethal double-strand breaks [49,50]. The comet assay results underscore the resilience of persisters in tolerating significant genomic damage while maintaining their viability and metabolic integrity.

Given this phenotypic complexity, we pursued targeted transcriptional analyses to dissect the molecular mechanisms orchestrating persister formation. Previous transcriptomic studies have established that persister cells typically exhibit decreased expression of genes involved in growth and metabolism and increased expression of stress-response and toxin–antitoxin (TA) systems [13,25,40,41]. Consistently, our RT-qPCR analyses revealed dynamic and selective induction of key persistence-related genes following pefloxacin exposure:(1)Proteolysis and toxin activation: The rapid induction of *clpP2* (but not *clpP1* or *lon*) protease suggests early targeted degradation of antitoxins, unleashing toxin activity, particularly *relE*, whose strong early expression (10 min) likely arrests translation and preserves energy in persister cells [41,51]. This is a novel finding, given the prior emphasis on *clpP1* as the primary protease in persistence mechanisms [52,53].(2)Protein repair and homeostasis: Chaperone genes (*clpC*, *clpX*) were strongly induced early, with sustained *clpX* induction at 30 min, suggesting ongoing proteostasis and repair of antibiotic-induced protein damage. Notably, *clpB*, which lacks direct association with proteolytic activity, remained largely unchanged, consistent with known Clp-chaperone interaction patterns [54,55,56].(3)Sequential toxin dynamics: We observed a distinct temporal expression of the toxins *relE* and *mazF*. While *relE* displayed a peak early in the persister enrichment process, *mazF* dominated at later stages (30 min), reinforcing growth arrest through mRNA degradation. This sequential activation reflects a coordinated regulatory response, fine-tuning persistence maintenance during prolonged antibiotic stress [57,58,59,60].(4)DNA damage and SOS response: *recA* showed rapid and substantial upregulation following pefloxacin exposure, consistent with its role in mediating the SOS response to DNA damage. Although necessary for persister survival under genotoxic conditions, *recA* induction likely represents a broader antibiotic stress response rather than a direct driver of persister formation *per se*. However, given that SOS activation can indirectly stabilize TA systems, its involvement cannot be entirely excluded [27,61,62].

Taken together, our data support a refined model (Appendix A) of pefloxacin-induced persister formation in *C. difficile*: Upon antibiotic exposure, rapid induction of *clpP2* (maybe associated with *clpX* and/or *clpc*) initiates the targeted proteolysis of antitoxins, activating TA systems (initially via *relE* toxin) and inducing immediate translational shutdown. Concurrently, extensive DNA damage stimulates the SOS response (*recA* activation). At intermediate and late stages, sustained chaperone induction (*clpX* and *clpC*) preserves protein homeostasis, while toxin dominance shifts toward *mazF*, reinforcing dormancy. Ultimately, these coordinated molecular events culminate in a robust persister phenotype characterized by metabolic quiescence, tolerance to extensive DNA damage, and antibiotic survival.

This study brings forward several novel contributions. First, we precisely delineate the temporal sequence of some gene expression changes underpinning the persister phenotype. The discovery that *clpP2*, rather than the previously implicated *clpP1*, drives proteolytic responses essential for toxin activation represents a fundamental advancement in understanding the molecular basis of persistence. Second, the identification of distinct temporal patterns of toxin (*relE* and *mazF*) induction (early dominance of *relE* followed by late induction of *mazF*) highlights a novel and nuanced toxin-specific regulatory strategy. Third, we clearly distinguish the dual role of the DNA-damage response (*recA*-mediated SOS pathway), demonstrating that although DNA damage is intrinsic to fluoroquinolone action, its activation, while necessary, does not directly dictate persistence but rather creates conditions favoring TA system stabilization and persister formation. The findings of this research significantly impact the current understanding of persistence in *C. difficile*. By elucidating specific molecular targets, such as ClpP2 peptidase, ClpX chaperone, and toxin components MazF and RelE, this study provides concrete, actionable insights that may lead to novel therapeutic strategies specifically designed to target persister populations. Consequently, our results might contribute to mitigating the clinical burden posed by recurrent CDI at the molecular level. In future studies, we expect to dilucidate how antibiotic persisters arise in vivo and how surviving populations contribute to virulence and antibiotic tolerance. Ultimately, such studies will help to improve therapeutic control of antibiotic–recalcitrant bacterial populations.

## 4. Materials and Methods

### 4.1. Bacterial Culture

*C. difficile* R20291 and Δ*spo0A* strains were grown in Brain and Heart Infusion Medium (BHIS) (BD Difco, San Jose, CA, USA) supplemented with 0.2% taurocholate, 0.1% fructose, and 0.1% glucose (all from Merck KGaA, Darmstadt, Germany). Cultures were grown at 37 °C under anaerobic conditions (H_2_ 8%, CO_2_ 5%, and N_2_ 87%) in a BACTRON EZ chamber (Shellab, Cornelius, NC, USA). To obtain exponential phase cultures, an overnight culture was performed, from which 1% *v*/*v* inoculum was added for an additional 5 h of growth.

### 4.2. Persistence Assay 

We started with a minimum inhibitory concentration (MIC) assay for pefloxacin, following the procedure previously described in [31]. Initial experiments were performed between 1x to 100x MIC. Then, exponential phase cultures were treated with 50-fold the MIC of pefloxacin (76.5 µg/mL) (Merck KGaA, Darmstadt, Germany) to induce the formation of persister cells. Subsequently, a persistence curve was performed by performing live cell counts at different post-treatment time intervals by serial dilutions, as described in [31]. At the end of each experiment, the bacteria were resuspended and subjected to the MIC assay again to verify that the results obtained were due to the formation of persister cells and not to the acquisition of antibiotic resistance.

### 4.3. Enrichment of Persister Cells

Exponential cultures treated with a 50-fold MIC concentration of pefloxacin (76.5 µg/mL) for 2 and 4 h were subjected to a lysis solution (1 mM NaOH, Merck, and 0.005% SDS, Merck) for 15 min as described in [31]. Aliquots were then inoculated at the appropriate times for counting colony forming units and determining survival.

### 4.4. Staining of Persister Cells

The culture enriched in persister cells induced by pefloxacin (76.5 µg/mL) was washed twice with TE buffer (10 mM Tris, 0.1 mM EDTA). To this culture, propidium iodide 40 ng/mL (Sigma-Aldrich, St. Louis, MI, USA) and thioflavin-T 5 µg/mL (Thermoscientific, Waltham, MA, USA) were added and incubated for 15 min in complete darkness. Thioflavin-T stained only bacteria with metabolically active RNA but not lysed bacteria [63,64]. Two additional washes with TE buffer were performed after the incubation period. Subsequently, the washed and stained cell suspension was observed using an Olympus™ epifluorescence microscope (Tokyo, Japan) equipped with FITC (fluorescein isothiocyanate with a center wavelength of 475 nm and FWHM of 35 nm) and TRITC (tetramethylrhodamine isothiocyanate with a center wavelength of 542 and FWHM of 20 nm) filters and a Qimaging Retiga 6 camera.

### 4.5. Flow Cytometry

Flow cytometry was performed using the BD FACSymphony™ A1 cell analyzer (BD Biosciences, San Jose, CA, USA). For each sample, unstained cultures containing 1% dimethyl sulfoxide (DMSO, Sigma-Aldrich) were used to determine the autofluorescence signal, considering size (FSC), complexity (SSC), and fluorescence. Cell viability was determined with propidium iodide (Sigma-Aldrich, 0.2 µg/mL), while metabolic activity was measured with thioflavin-T (Thermoscientific, Waltham, MA, USA, 30 µg/mL). A total of 100,000 events were analyzed unless otherwise noted. Data were analyzed using FlowJo software (version 10.7.2, FlowJo LLC, Ashland, Jackson County, OR, USA).

### 4.6. Comet Assay

DNA damage generated by pefloxacin in persister cells was evaluated as described in Solanky & Haydel 2012 [38]. Cultures were treated with pefloxacin at different times and with lysis buffer for 15 min. Subsequently, a layered microgel was formed on a microscope slide by adding 5 µL of the culture as described previously [38]. The gel was then exposed to a cell lysis solution (1% Triton X-100, 2.5 M NaCl, 100 mM EDTA, 10 mM Tris 10 mM pH 10, SDS 1%, lysozyme 10 mg/mL, and proteinase K 20 mg/mL) for 1 h, followed by an enzyme lysis solution (2.5 mM NaCl, 10 mM EDTA, 10 mM Tris pH 10, and proteinase K 1 mg/mL) for 2 h at 37 °C, all in the dark. The gel was transferred to an electrophoresis chamber containing 300 mM sodium acetate and 100 mM Tris pH 9 and subjected to electrophoresis at 22 V for 40 min. Finally, the gel was immersed in 1 M ammonium acetate solution, followed by absolute ethanol and 70% ethanol, then stained with propidium iodide (Sigma-Aldrich, 40 ng/mL) and observed at 400x total magnification using an Olympus™ epifluorescence microscope. Images were taken from 5 different fields using the Qimaging Retiga 6 camera, and comet length was then measured using ImageJ software.

### 4.7. RNA Extraction and Real-Time Quantitative PCR

For the relative quantification of transcript levels of persister cells enriched in lysis buffer, antibiotic treatment was performed for 10 and 30 min in the exponential phase. RNA extraction was performed using the acid phenol method as described in [65]. Subsequently, 1 µg of total RNA was converted into cDNA using reverse transcriptase and random primers (Promega, Madison, WI, USA). These cDNAs were used for qRT-PCR as follows: a 10 µL reaction containing 25 ng cDNA, 5 µL Brilliant II SYBR Green QPCR Master Mix (Agilent Technologies, Santa Clara, CA, USA), 0.25 µM of each primer (Appendix A), and water. PCR conditions were 50 °C for 2 min, 95 °C for 10 min, followed by 95 °C for 10 s, 60 °C for 30 s, and 72 °C for 30 s for a total of 40 cycles. Melting curves were generated by increasing the temperature by 1 °C within 60 °C to 95 °C. Real-time PCR data were analyzed using 16s rRNA and *dnaK* levels for normalization according to a previously detailed methodology [65]. Results were plotted using GraphPad Prism 7 software. The experiment was performed with three biological replicates, each with three technical replicates.

### 4.8. Statistical Analysis

Prism GraphPad 9.0 software (BOSTON, MA 02110, USA) was used to graph and analyze the data using statistical tests for nonparametric data, such as one-way and two-way ANOVA, accompanied by Sidak and Bonferroni post hoc tests.

## Figures and Tables

**Figure 1 antibiotics-14-00628-f001:**
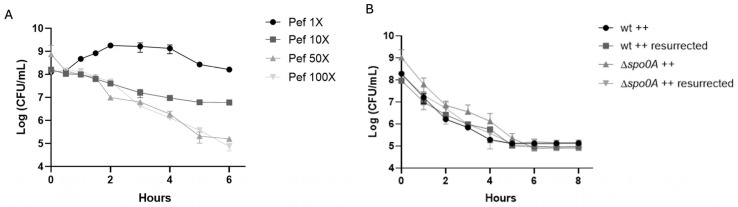
Pefloxacin-induced persistence in *C. difficile* R20291. (**A**) Time–kill kinetics of exponential-phase cultures treated with pefloxacin at 1×, 10×, 50×, and 100× the determined MIC (1.53 µg/mL). High-dose treatments (50× MIC only) display a characteristic biphasic killing curve: an initial rapid decline in viable counts followed by a persistent plateau (*n* = 3; mean ± SD). (**B**) Rechallenge of survivors from the 50× MIC treatment and comparison of wild-type (wt) versus non-sporulating Δ*spo0A* mutant under identical conditions. Cultures were washed, resuspended in BHIS++ (supplemented with taurocholate, fructose, and glucose), and exposed again to 50× MIC pefloxacin (76.5 µg/mL). The reproduced biphasic kill profile and unchanged MIC confirm a reversible, non-sporulating persister phenotype (*n* = 3; mean ± SD).

**Figure 2 antibiotics-14-00628-f002:**
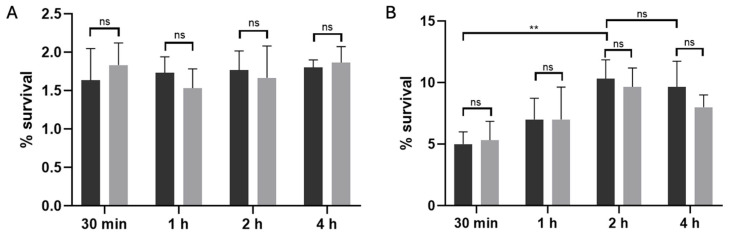
Lysis-enriched persister cell survival in *C. difficile* R20291. (**A**) Basal persister frequency following lysis without antibiotic exposure. Wild-type (black bars) and Δ*spo0A* (gray bars) cultures were treated with lysis buffer alone, yielding a low, time-independent survival of ~1.8% (*n* = 3; mean ± SD), consistent with stochastic (type I) persister formation. (**B**) Antibiotic-induced persister enrichment. Cultures pre-treated with 50× MIC pefloxacin (76.5 µg/mL) for 30 min, 1, 2, and 4 h were washed, subjected to the same 15 min lysis, and plated for CFU. Survival increased in a time-dependent manner, peaking at ~10% after 2 h and remaining stable through 4 h (*n* = 3; mean ± SD). No significant difference was observed between wt and Δ*spo0A* at any time point (ns), indicating that persister enrichment is independent of *Spo0A*-mediated sporulation. (** *p* < 0.005).

**Figure 3 antibiotics-14-00628-f003:**
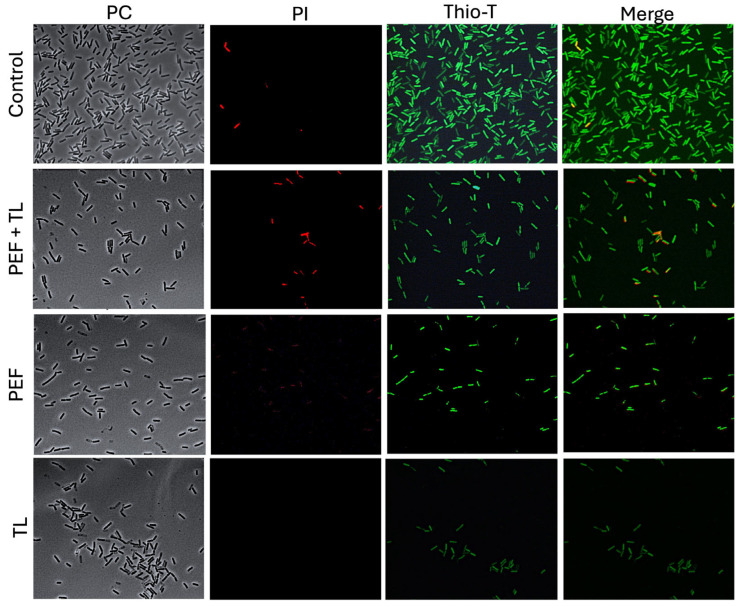
Fluorescence assessment of metabolic activity and membrane integrity in persister-enriched *C. difficile* R20291. Wild-type R20291 cultures in exponential phase were subjected to one of four conditions: no treatment (Control), lysis buffer only (TL), pefloxacin at 50× MIC only (PEF), or pefloxacin followed by lysis (PEF + TL). Cells were (**left**) imaged by phase-contrast (CF), (**middle**) stained with PI (red) to reveal membrane permeability, and (**right**) stained with Thio-T (green) to report RNA metabolic activity (*n* = 3).

**Figure 4 antibiotics-14-00628-f004:**
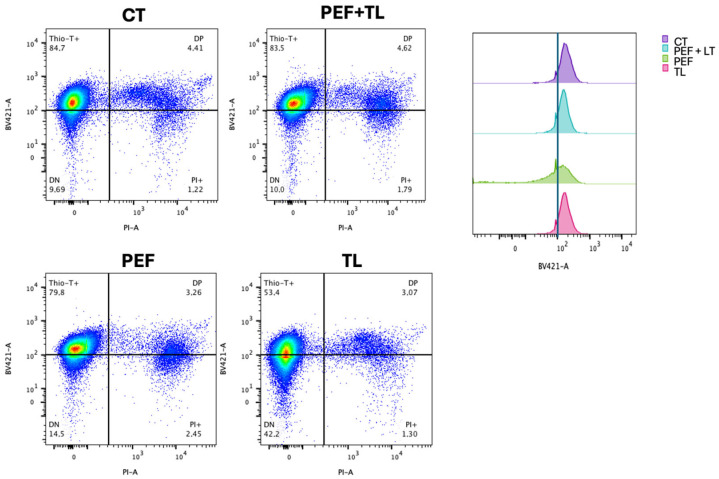
Flow cytometric profiling of *C. difficile* persister induction and enrichment. (**Left**) Representative dual-parameter dot plots of exponential-phase cultures: untreated control (CT), pefloxacin alone (PEF), lysis treatment alone (TL), and combined pefloxacin + lysis (PEF + TL). Cells were stained with Thio-T (metabolic activity) and PI (membrane integrity). Quadrants delineate Thio-T^+^ only (**upper left**), PI^+^ only (**lower right**), double-negative (DN; **lower left**) and double-positive (DP; **upper right**) subpopulations, with percentages indicated. (**Right**) Overlaid histograms of Thio-T fluorescence for each condition. The dashed vertical line marks the threshold separating Thio-T^−^ and Thio-T^+^ signals.

**Figure 5 antibiotics-14-00628-f005:**
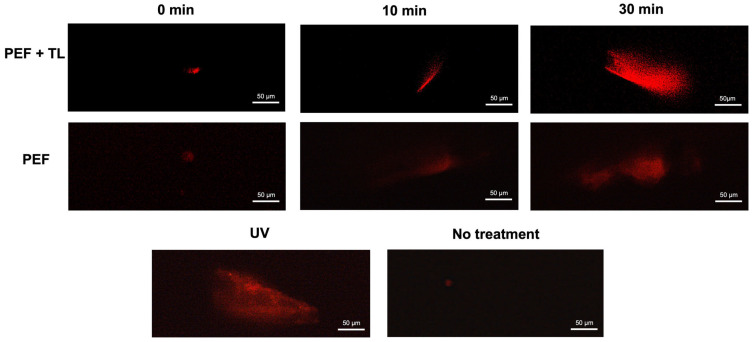
DNA degradation in pefloxacin-induced *C. difficile* enriched persister cells. Comet assay performed on persister cells enriched by lysis treatment and previously exposed to pefloxacin 50× (**top panel**) and cells exposed to pefloxacin 50× without lysis treatment (**middle panel**) for 0, 15 and 30 min. Untreated cells were used as negative control and cells treated with UV radiation for 10 min were used as positive control (**lower panel**) (*n* = 3). Images of 5 fields were taken and the most representative was selected.

**Figure 6 antibiotics-14-00628-f006:**
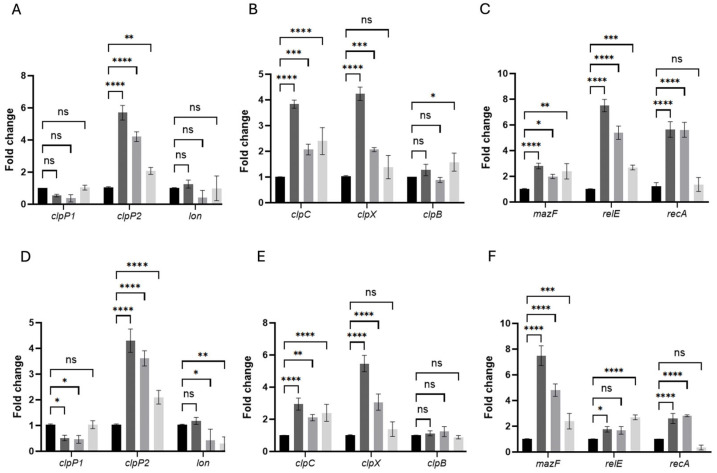
Persistence-associated genes are rapidly induced in pefloxacin-enriched *C. difficile* persister cells. RNA was extracted at 10 min (**A**–**C**) and 30 min (**D**–**F**) after treatment with 50× MIC pefloxacin following lysis enrichment. Transcript levels of the peptidase genes *clpP1*, *clpP2* and the *lon* protease (**A**,**D**), the chaperone genes *clpC*, *clpX* and *clpB* (**B**,**E**), and the toxin–antitoxin components *mazF*, *relE* as well as the SOS regulator *recA* (**C**,**F**) were measured by RT-qPCR and normalized to 16S rRNA. Data are presented as mean ± SD of three biological replicates (*n* = 3), with statistical significance relative to untreated controls indicated as * *p* < 0.01, ** *p* < 0.005, *** *p* < 0.001, **** *p* < 0.0001, and ns (not significant). Black bars, untreated control (CT); dark-gray bars, pefloxacin + lysis (PEF + TL); medium-gray bars, pefloxacin alone (PEF); light-gray bars, lysis alone (TL).

## Data Availability

The original contributions presented in this study are included in the article/Appendix A. Further inquiries can be directed to the corresponding author(s).

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
