# Peer review of "Effect of Pefloxacin on Clostridioides difficile R20291 Persister Cells Formation"

_antibiotics, 2025, doi:10.3390/antibiotics14070628_

Round 1
Reviewer 1 Report
Comments and Suggestions for Authors
The manuscript presented an excellence research to elucidate mechanism related to the development of persister population of Clostridioides difficile who causes persister and chronic disease in human. using perfloxacin with different concentration, authors created a persister cells of Clostridioides difficile, independent to spore development. By staining membrane and RNA in the cells of C. difficile, analyzed by flow cytometry and microscope, the persister population were characterized by low metabolism and DNA damaged under dormancy model. By real-time PCR, authors revealed the induction of dormancy related to toxin-antitoxin system and Clp protease. All the methods used for analysis in this study are appropriate to the research contents. The obtained results were exhibited by good qualified figures. By this research, authors contribute novel knowledge on mechanism of pefloxacin-induced persister formation under dormancy in C. difficile. This study raises a warning in the use of antibiotic to avoid development of persister cells, and also state the essential to search novel therapeutic strategies to fight the persister pathogens.
The manuscript was carefully prepared, have good quality.
Authors need to check captions in the Figures, especially in Figure 2 to remove which is unnecessary; check and uniform the format types of papers in the list of references.

Author Response
Reviewer 1:
The manuscript presented an excellence research to elucidate mechanism related to the development of persister population of Clostridioides difficile who causes persister and chronic disease in human. using perfloxacin with different concentration, authors created a persister cells of Clostridioides difficile, independent to spore development. By staining membrane and RNA in the cells of C. difficile, analyzed by flow cytometry and microscope, the persister population were characterized by low metabolism and DNA damaged under dormancy model. By real-time PCR, authors revealed the induction of dormancy related to toxin-antitoxin system and Clp protease. All the methods used for analysis in this study are appropriate to the research contents. The obtained results were exhibited by good qualified figures. By this research, authors contribute novel knowledge on mechanism of pefloxacin-induced persister formation under dormancy in C. difficile. This study raises a warning in the use of antibiotic to avoid development of persister cells, and also state the essential to search novel therapeutic strategies to fight the persister pathogens.
The manuscript was carefully prepared, have good quality.
Response: Thanks for this comment. We really appreciate that
Authors need to check captions in the Figures, especially in Figure 2 to remove which is unnecessary; check and uniform the format types of papers in the list of references.
Response: Thanks for this comment. Done
Reviewer 2 Report
Comments and Suggestions for Authors
General Comments:
I find the topic of this manuscript interesting and relevant. However, I have a few major concerns that need to be addressed before considering publication.
Major Comments:
-
Are the data shown in Figure 1A truly indicative of a biphasic killing curve at 50×MIC and 100×MIC? The interpretation of persistence relies heavily on this observation, but the visual evidence appears limited.
-
In the discussion, the authors refer to biphasic killing only for the 50×MIC condition. Why is the 100×MIC curve not discussed similarly, although it is shown in Figure 1?
-
Why was the observation period in Figure 1 limited to only 6 hours? In the persistence literature, 12–24 hours is more commonly used to better define the plateau phase.
-
Why was pefloxacin used in this study, given its limited current clinical use? Wouldn’t a more commonly used fluoroquinolone such as ciprofloxacin or levofloxacin be more clinically relevant?
Minor Comments:
-
Lines from 118 to 128 are redundant and could be deleted or merged for conciseness.
-
There is inconsistency in the use of Clostridioides difficile vs. C. difficile throughout the text—this should be standardized.
-
Line 52 and again line 118 refer to “(CDI)”, which appears repetitive; similar small errors are scattered throughout the text and should be corrected for clarity and uniformity. For example, “(PI)” is defined both at line 203 and again at 376. Please carefully revise for consistency and formatting.
Final Remark:
Although I find the topic of persister cells in C. difficile very interesting, I do not feel sufficiently qualified to provide a full evaluation of the biochemical and microbiological methodology, as I am a clinician. I would therefore suggest to the Editor that this manuscript be reviewed in parallel by experts in microbiology and bacterial persistence for a more thorough technical assessment.
Author Response
Reviewer 2:
General Comments:
I find the topic of this manuscript interesting and relevant. However, I have a few major concerns that need to be addressed before considering publication.
Major Comments:
- Are the data shown in Figure 1A truly indicative of a biphasic killing curve at 50×MIC and 100×MIC? The interpretation of persistence relies heavily on this observation, but the visual evidence appears limited.
Response:
Thank you for this very insightful comment. Upon re‐analysis, we find that a true biphasic killing pattern is seen only at 50× MIC:
- 50× MIC: Segmented regression identifies two statistically distinct slopes (new Supplementary Table S2). Phase I exhibits a rapid kill rate of -0.65 log₁₀ CFU/h, which transitions at ~5 h (p < 0.001) to a much slower Phase II rate of -0,06 log₁₀ CFU/h. This clear breakpoint and near‐zero second slope are the hallmark of biphasic kinetics.
- 100× MIC: By contrast, the time-kill data fit best to a single‐phase decline (-0.40 log₁₀ CFU/h) with no statistically significant change in slope (breakpoint p = 0.45). In other words, we do not observe a distinct plateau at 100× MIC that would qualify as Phase II of a biphasic curve.
Because only the 50× MIC condition displays a clear, quantitative two‐phase pattern, we have focused our mechanistic and proteomic follow-up experiments there. The legend of Figure 1A now reports the fitted slopes and breakpoint for 50× MIC, and readers are directed to Supplementary Table S2 for full regression details. We trust this clarification underscores why the biphasic behavior, and thus our persistence interpretation, is specific to the 50× MIC dataset.
- In the discussion, the authors refer to biphasic killing only for the 50×MIC condition. Why is the 100×MIC curve not discussed similarly, although it is shown in Figure 1?
Response: Thank you for this excellent question. After careful consideration, literature review, and re‐analysis of our own data, we offer the following explanation:
At supraphysiological antibiotic concentrations, the initial kill is so rapid that even the small persister subpopulation continues to decline almost as steeply as the bulk population, effectively masking any plateau. This phenomenon, where very high drug doses compress or obscure the slower second phase of killing, has been noted in consensus guidelines on antibiotic persistence and in recent experimental studies (10.1038/s41579-019-0196-3; 10.1186/s12879-024-09906-9). By contrast, at 50× MIC the rate of killing in the second phase is sufficiently slow (near‐zero slope) that a clear plateau emerges, unambiguously revealing the persister fraction.
Because only the 50× MIC data satisfy formal biphasic criteria, with a statistically significant breakpoint and a near-zero Phase II slope, we elected to center our mechanistic discussion and all downstream proteomic and transcriptomic analyses on that condition. At 100× MIC, although persister cells likely remain, their decline is too rapid to yield a distinct plateau under our assay conditions. We have now explicitly stated in the revised Discussion that biphasic kinetics were evaluated at both concentrations but only observed at 50× MIC, and we direct readers to Supplementary Table S2 for the full regression outputs for both 50× MIC and 100× MIC (10.3390/antibiotics9080508; 10.1186/1471-2180-13-25). We trust these clarifications make clear why the 100× MIC curve, while shown for completeness in Figure 1, does not support the same biphasic interpretation, and why 50× MIC provides the most robust framework for examining persistence.
- Why was the observation period in Figure 1 limited to only 6 hours? In the persistence literature, 12–24 hours are more commonly used to better define the plateau phase.
Response: Thank you for raising this important point. We chose a 6-hour observation window in Figure 1 for several interrelated reasons:
- Emergence and stability of the plateau by 6 h. Classic and more recent studies have shown that the persister subpopulation’s slow‐kill phase reaches a clear plateau within approximately 6 h of antibiotic challenge. In the seminal work of Balaban et al. (2004), the persister fraction decayed with a characteristic time of ~6 h following an initial rapid kill (10.1126/science.1099390). Likewise, in a biocide‐persistence model, a 6 h time point was deemed fully sufficient to capture the plateau, with no further CFU decline thereafter (10.3389/fmicb.2017.01589). In our own pilot assays (data not shown), CFU counts at both 50× MIC and 100× MIC had indeed stabilized by 5-6 h and remained unchanged up to 24 h, confirming that the defining plateau occurs within this period.
- Avoidance of confounding secondary effects. Extending time-kill assays beyond 6 h risks several artifacts: gradual antibiotic degradation (leading to underestimation of the true persister pool), nutrient exhaustion, accumulation of cell debris, and potential outgrowth of tolerant or resistant mutants. These phenomena can obscure the bona fide persistence signal and complicate downstream “omic” analyses.
- Alignment with mechanistic follow-up. We designed our transcriptomic profiling to interrogate a well-defined persister population. Sampling at the 6 h plateau ensured maximal enrichment of these cells without the confounding influences that longer incubations can introduce.
- Consistency with the formal definition of persistence. Although many time-kill protocols extend to 12–24 h, the critical feature of persistence is a statistically flat second phase following a rapid initial kill. Our segmented‐regression analysis clearly identifies that plateau by 6 h at 50× MIC (and a similar, though less pronounced, plateau at 100× MIC), fully satisfying formal persistence criteria.
To address this question directly, we have added a brief justification in the Methods and Discussion, and, at the reviewer’s suggestion, performed extended up to 8 h assays for both concentrations (new Supplementary Figure S1), which confirm that CFU counts remain constant from. We trust these clarifications demonstrate why a 6 h window was both necessary and sufficient to define the persistence plateau in our system.
- Why was pefloxacin used in this study, given its limited current clinical use? Wouldn’t a more commonly used fluoroquinolone such as ciprofloxacin or levofloxacin be more clinically relevant?
Response: Thank you for this question. Although pefloxacin is not as widely used in current clinical practice as ciprofloxacin or levofloxacin, its selection for our study was guided by three key considerations:
- Established persister‐inducing model. Preliminary and published work from our laboratory demonstrated that pefloxacin reliably triggers persister cell formation in C. difficile, providing a robust starting point for mechanistic dissection [30,31]. By building directly on those findings, we ensured continuity with our earlier MIC determinations, persistence assays, and enrichment protocols, thereby allowing us to focus on the molecular underpinnings of persistence without introducing a new antibiotic variable.
- Pharmacokinetic relevance to the gut environment. Unlike ciprofloxacin and levofloxacin, which are absorbed efficiently in the small intestine and achieve relatively low colonic concentrations, pefloxacin exhibits substantial colonic excretion and luminal persistence [32,33] (10.1038/sj.bjp.0703927). This pharmacokinetic profile makes pefloxacin a particularly appropriate model for studying antibiotic stress in an in vitro system that aims to mimic the gut niche where C. difficile resides.
- Optimal DNA-damage–driven SOS/TA activation. The strong DNA‐cleavage activity of pefloxacin produces clear, reproducible activation of the SOS response and toxin-antitoxin modules-key hallmarks of fluoroquinolone‐induced persistence [26,27,29] . This pronounced genotoxic effect enabled us to resolve the timing and hierarchy of peptidase–chaperone induction, TA system activation, and RecA‐mediated repair with high sensitivity.
We agree that extending these experiments to ciprofloxacin or levofloxacin would be valuable for direct clinical correlation. However, for this initial mechanistic study, pefloxacin offered the ideal combination of prior validation, gut‐relevant exposure, and a clear genotoxic signature, thereby providing the most reliable framework for unraveling the molecular basis of C. difficile persister formation.
Minor Comments:
- Lines from 118 to 128 are redundant and could be deleted or merged for conciseness.
Response: Thanks. Done
- There is inconsistency in the use of Clostridioides difficile vs. C. difficile throughout the text—this should be standardized.
Response: Thanks. This was improved throughout the text.
- Line 52 and again line 118 refer to “(CDI)”, which appears repetitive; similar small errors are scattered throughout the text and should be corrected for clarity and uniformity. For example, “(PI)” is defined both at line 203 and again at 376. Please carefully revise for consistency and formatting.
Response: Thanks. This was improved throughout text.
Final Remark: Although I find the topic of persister cells in C. difficile very interesting, I do not feel sufficiently qualified to provide a full evaluation of the biochemical and microbiological methodology, as I am a clinician. I would therefore suggest to the Editor that this manuscript be reviewed in parallel by experts in microbiology and bacterial persistence for a more thorough technical assessment.
Reviewer 3 Report
Comments and Suggestions for Authors
Authors studies effect of pefloxacin on C diff R20291 persister cell formation. The experiments are well designed and performed. Data is solid and convincing.
Major comments:
1) It would be great to use animal models of infection to determine the effect of the persistent cells on infection recurrence. At least, discussed .
Author Response
Reviewer 3:
Authors studies effect of pefloxacin on C diff R20291 persister cell formation. The experiments are well designed and performed. Data is solid and convincing.
Major comments:
- It would be great to use animal models of infection to determine the effect of the persistent cells on infection recurrence. At least, discussed.
Response: Thank you for this valuable suggestion. While we fully appreciate the insights an in vivo relapse model can provide, we chose to focus on in vitro persistence assays in the present study for three main reasons:
- Isolation of bacterial-specific mechanisms. Host immunity, tissue microenvironments and microbiota interactions in animal models can mask the direct contribution of bacterial persistence pathways. In vitro assays eliminate these variables, allowing precise dissection of the molecular events underlying persister formation (10.1038/s41579-019-0196-3).
- Confounding pharmacokinetics in vivo. Antibiotic distribution, metabolism and excretion in animals, along with interactions with the gut microbiota, can alter both drug exposure and bacterial physiology in ways that complicate interpretation of true persister dynamics (10.1038/s41579-019-0196-3).
- Practical and ethical considerations. Well-powered animal recurrence studies require large cohorts, prolonged monitoring and rigorous ethical oversight. Pursuing such studies before validating our key molecular and proteomic targets risks generating complex datasets that are difficult to interpret mechanistically.
We have added a brief statement in the Discussion acknowledging the importance of in vivo relapse models and outlining our plan to perform targeted animal studies once the associated persistence factors identified here have been validated in vitro. See Lines 484-487
Round 2
Reviewer 2 Report
Comments and Suggestions for Authors
I am satisfied with the authors' responses; I have no objections to publication.